# Intelligent Device for Harvesting the Vibration Energy of the Automobile Exhaust with a Piezoelectric Generator

**DOI:** 10.3390/mi14020491

**Published:** 2023-02-20

**Authors:** Jie Huang, Cheng Xu, Nan Ma, Qinghui Zhou, Zhaohua Ji, Chunxia Jia, Shan Xiao, Peng Wang

**Affiliations:** 1Beijing Information Technology College, Beijing 100015, China; 2Beijing Key Laboratory of Information Service Engineering, Beijing Union University, Beijing 100101, China; 3Beijing University of Technology, Beijing 100124, China; 4Beijing University of Civil Engineering and Architecture, Beijing 100044, China

**Keywords:** piezoelectric generator, self-powered intelligent device, automobile exhaust system, vibration energy, signal monitoring

## Abstract

With increasing consumption of energy and increasing environmental pollution, research on capturing the vibration energy lost during transportation and vehicle driving is growing rapidly. There is a large amount of vibration energy in the automobile exhaust system that can be recycled. This paper proposes a self-powered intelligent device (SPID) using a piezoelectric energy generator. The SPID includes a piezoelectric generator and sensor unit, and the generator is installed at the end of the automobile exhaust system. The generator adopts a parallel structure of four piezoelectric power generation units, and the sensing unit comprises light-emitting diode warning lights or low-power sensors. A simulated excitation experiment verifies the working state and peak power of the piezoelectric generator unit, which can achieve 23.4 μW peak power. The self-power supply and signal monitoring functions of the intelligent device are verified in experiments conducted for driving light-emitting diode lights and low-power sensors. The device is expected to play a crucial role in the field of intelligent driving and automobile intelligence.

## 1. Introduction

The energy crisis relating to oil shortages and environmental pollution, such as air pollution, have introduced great challenges to the automobile industry [1,2]. As a result, research on capturing the energy lost during vehicle transportation is growing rapidly [3,4,5]. The technology used in collecting the energy loss of each part of the automobile body is important to the development of the automobile industry in that it improves the energy efficiency and fuel economy of automobiles. In the process of driving, the released heat, braking energy, and vibration energy are the main targets of energy collection technology [6,7,8,9]. As an important part of the automobile chassis, the exhaust system produces a lot of vibration because it is connected to the engine and must exhaust the exhaust gas [10]. The vibration energy can be collected for electronic equipment detection and driving sensors and even more promising applications [11,12,13]. Additionally, in the past 10 years, there has been much research on energy collection for low-power power generation, ultimately for energy recovery or the self-power supply of monitoring elements [14,15,16,17,18,19,20]. Such energy collection poses a good solution for the replacement and maintenance of the battery in a sensor element, thus reducing the maintenance cost of vehicle sensor equipment and even various types of engineering equipment [21,22,23,24,25,26,27].

In the current research, piezoelectric power generation is a stable traditional microscale energy capture technology. Piezoelectric materials have a good electromechanical coupling effect and high energy conversion efficiency. The most well-known piezoelectric materials are piezoelectric ceramics and piezoelectric polymers [24,28,29,30,31]. They have the advantage of converting energy harvested from the environment into electrical energy, which is used to directly or indirectly drive electronic devices [32,33]. For example, piezoelectric materials collect the vibration energy transmitted to the ground when a person walks, the energy generated by friction between a person and a backpack, the vibration energy generated by wind, the longitudinal transverse waves of the sea, and the vibration energy generated by high-rise buildings [34,35,36,37]. To improve the performance of piezoelectric materials, new composite materials, and new structure designs have been put forth by several academics [38,39,40,41]. Furthermore, piezoelectric materials are widely used in the aerospace and automotive industries [42,43,44]. Lafarge et al. used piezoelectric energy harvesters to collect the vibration energy inside vehicle suspension systems and tires, and the harvested energy can be used to power embedded wireless sensors [45]. The existing technology demonstrates that the vibration energy harvester has been widely used to collect vibration energy in various environments [46,47]. The automobile exhaust system has a large area of vibration [48,49]. If the vibration energy can be collected, it can be used in vibration energy harvesting automobile intelligent sensing and condition monitoring.

In view of little research on mechanical vibration energy harvesting of the exhaust system, this paper proposes a new self-powered intelligent device (SPID) using a piezoelectric energy generator. Taking a light vehicle as the research object, the SPID is installed at the end of the exhaust system. The system adopts a parallel structure of four piezoelectric power generation units, including light-emitting diode (LED) warning units or low-power sensor units. By conducting simulation excitation experiments for different amplitudes and frequencies, the working state and peak power of the piezoelectric power generation unit are verified. The piezoelectric power generation unit produces 23.4 μW peak power. A simulation model of a piezoelectric chip is established and the results are compared with test results. The operational ability of the SPID is verified in experiments for an LED lamp and low-power sensor. The SPID effectively realizes the recovery of vibration energy of automobile exhaust devices and the self-energy supply of sensors and signal monitoring functions. This device is expected to play a crucial role in the field of intelligent driving and automobile intelligence [50,51].

## 2. Results and Discussion

### 2.1. Structural Design and Working Principle

The components of the self-powered intelligent device (SPID) are a base, piezoelectric plate, exhaust pipe, splint, sleeve, and sensor (Figure 1c). The integral device is installed at the end of the automobile exhaust pipe (Figure 1a,b). The base plate is installed on the vehicle chassis, the four piezoelectric pieces are installed around the base plate, the sleeve is divided into two parts and installed on the exhaust pipe, and the clamp plate is installed around the sleeve. Piezoelectric sheets of three sizes are used. The piezoelectric zirconate titanate (PZT) materials are evenly distributed on each piezoelectric sheet and connected to applications such as sensors or warning lights through wires (Figure 1d).

Regarding the working principle of the SPID (Figure 2), the piezoelectric pieces are distributed above and on both sides of the exhaust pipe. The PZT material is uniformly distributed on the piezoelectric sheet. According to the principle of the generation of piezoelectric power and the principle of electronic polarization, when the piezoelectric sheet is deformed, polarization charges appear on the surface of the PZT material. The two piezoelectric pieces are distributed above the exhaust pipe. When the exhaust pipe shakes from left to right, the piezoelectric pieces undergo elastic deformation. The other two piezoelectric pieces are distributed on either side of the exhaust pipe. When the exhaust pipe vibrates up and down, the piezoelectric pieces undergo elastic deformation (Figure 2a). The piezoelectric piece can slide between the splints without impeding the vibration in the other direction. When the piezoelectric sheet deforms, the PZT on its substrate deforms. After the piezoelectric material deforms, its internal electrons are polarized, and free electrons transfer to one side of the PZT, such that a current flows in the external circuit. When the PZT deforms in the opposite direction, the internal electrons are reverse polarized and move in the opposite direction. The external circuit forms a reverse current (Figure 2b). The exhaust pipe drives the piezoelectric piece to swing up and down and left and right, and the PZT is internally polarized many times. Free electrons move to both ends of the PZT many times. The external circuit generates a repeated current. Additionally, the stress under two deformation states of the PZT is simulated utilizing simulation software. The stress distribution trend of the PZT in different states shows that the electrons in the PZT are moving in the deformed state. The potential difference drives electrons to form an electric current in the external circuit.

### 2.2. Output Performance

In the study of the electrical output performance of the SPID, the linear motor drives the base to move, and the performances of piezoelectric pieces with different specifications are tested. The piezoelectric pieces are installed on the stator base, and clamping plates are installed on the movable base. The linear motor drives the movable pieces in a straight line. The experiment simulates a linear vibration environment. The overall layout of the experiment is shown in Appendix A. The frequency of the linear motor is set at 1 Hz, and the excitation amplitude is 1 mm. The output performance of PZT units 1, 2, and 3 (PZT-1, PZT-2, and PZT-3) are tested for different specifications. Because the PZT is evenly distributed on the two sides of the piezoelectric sheet, a performance comparison experiment of the PZT on both sides is carried out to select the best PZT power generation unit. The voltage performances of sides A and B of PZT-1, PZT-2, and PZT-3 are tested for a motor stroke of 1 mm (Figure 3a,b) and a motor stroke of 2 mm (Figure 3d,e). A photograph of linear electrode travel of 1 and 2 mm is shown in Figure 3c. It is seen from the experimental data that the performance of side A is better for the PZT units of different sizes at the different motor strokes. Side A and a stroke of 2 mm are thus selected in subsequent experiments. In addition, under the same excitation stroke, PZT-3 has local plastic deformation at the fixed end due to the excessive force of the fixing screw, and The performance of power generation is somewhat decreased. Therefore, PZT-2 has the best voltage performance. To better compare the performances of PZT-1A, 2A, and 3A, the power performance of each PZT is tested by means of series resistance. The comparison experiment shows that the power of PZT-2A can reach 6.6 μW, which is the highest power among the three specifications of PZT. Therefore, under the current test conditions, PZT-2A is the best power generation unit (Figure 3f).

Experiments with different strokes and different excitation frequencies are designed to further verify the performance of PZT units having different specifications. First, the effects of the excitation stroke on the power generation performance of the PZT are verified. The linear motor is set to have strokes of 1, 2, 3, 4, and 5 mm, and excitation tests are conducted for PZT units having different specifications. The output voltage of the PZT units having different specifications increases with the loading stroke (Figure 4a–c). A comparison of the above data reveals that PZT-2 has the best voltage performance. At a stroke of 5 mm, the voltage reaches 22 V. The current and power performances for the different specifications of PZT are shown in Appendix A. The vibration amplitude of the automobile exhaust funnel generally ranges from 1 to 10 mm. This experiment verifies that the deformation degree of PZT is suitable for collecting the vibration energy of the automobile exhaust funnel. Furthermore, to verify the response of PZT piezoelectric sheets to the excitation frequency, voltage performance comparison experiments for the various specifications of PZT under different vibration frequencies are designed. A photograph of the test bench is shown in Appendix A. The test bench uses a rotor motor to achieve high-frequency vibration. The output frequency of the vibration platform can reach 1, 5, 10, 20, 30, 40, and 50 Hz, and the amplitude is 1–5 mm. The vibration environment of the exhaust system can be simulated under the common speed range of a four-cylinder vehicle [11]. The voltage data of the different specifications of PZT are obtained under these experimental conditions (Figure 4d–f). The PZT of each specification has good power generation performance at different frequencies, which demonstrates that PZT can be applied to the vibration environment of an exhaust system. Figure 4d–f shows that the PZT generates different voltages according to the magnitude of the random amplitude. The current and power are shown in Appendix A for different specifications of PZT. The comprehensive performance evaluation shows that PZT-2 has the best voltage, current, and power performance, and PZT-2 is thus selected as the experimental object for subsequent experiments.

A scheme of multiple PZT power generation units working at the same time is designed to collect more vibration energy from the automobile exhaust system. Through the reasonable design of the layout and structure, four power generation units are installed around the device. The performances of PZT power generation units in series and parallel are tested, and the maximum power output mode is found. The circuit connection modes of the PZT units in parallel and series are shown in Appendix A. First, the power generation performance of PZT1–4 single units is shown in Appendix A. Figure 5a,b shows the voltage and current performance of different numbers of PZT units in parallel. With more parallel units, the output voltage remains unchanged, but the output current gradually increases. Figure 5c,d shows the voltage and current performance of different numbers of PZT units in series. A greater number of units in series increases the output voltage and the output current slightly. The power output is tested to better compare the performance output of parallel and series schemes. The power reaches 23.4 μW for four PZT units in parallel (Figure 5e). The output performance of four PZT units in series is 18.4 μW (Figure 5f). According to the performance comparison, the four PZT units output the maximum power in parallel. Therefore, in the application experiment, the parallel connection method is used for testing.

### 2.3. Demonstration

To reflect the functionality of PZT in the field of vehicle vibration energy collection, experiments are conducted for driving low-power consumption sensors and driving indicator lamps (Figure 6). Four PZT units are assembled on the linear motor in parallel, and a vibration environment is simulated using linear motion (Figure 6a). In the warning-light experiment, 35 LEDs are arranged into the letters BITC through the assembly of a circuit on a breadboard. The link mode is shown in Figure 6b. Driven by the linear motor, the LED warning light is successfully lit, as shown in Figure 6c. In further verifying the performance of the piezoelectric generator, commercial capacitors, and rectifier bridges are used to build the power supply circuit of small sensors. The piezoelectric generator easily drives the commercial thermometer and hygrometer by charging the capacitor (Figure 6d). The output performance of the PZT unit is very stable after 430,000 cycles (Figure 6e). The intelligent device is installed on a light truck. When the truck starts, the vibration energy at the end of the exhaust pipe is collected by the intelligent device, and the LED warning light is driven to demonstrate the feasibility of the intelligent device (Figure 6f,g). Additionally, the signal generated by the piezoelectric generator can be used to monitor the vibration of equipment. The device can be installed on the vehicle-mounted air pump to monitor its working frequency. When the onboard air pump starts or is disturbed, vibration signals with different frequencies and amplitudes appear (Figure 6h,i). More information is presented in a Appendix A. All the above application experiments demonstrate the feasibility of the intelligent device to capture vibration energy.

## 3. Experimental

### 3.1. Fabrication of the SSAS

The diameter and height of the self-powered intelligent device (SPID) are 210 mm and 110 mm, respectively. The whole device is installed at the end of the automobile exhaust pipe, wherein the base plate is installed on the automobile chassis, and four piezoelectric pieces are respectively installed around. The sleeve is divided into two parts and installed on the exhaust pipe, and the clamp plate is installed around the sleeve. Three sizes of piezoelectric sheets are used. PZT materials are evenly distributed on each piezoelectric sheet, which are connected to applications such as electrical appliances or warning lights through wires. The base plate sleeves are all 3D printed, and the material is polylactic acid (PLA). The PZT-2 has a thickness of 80 μm. its length and width are 60 mm and 30 mm, respectively. The thickness of the splint is 50 mm; its length and width are 20 mm and 10 mm, respectively.

### 3.2. Electrical Measurement

The self-powered intelligent device (SPID) is driven by a linear motor (CP-1100, LinMot, Lake Geneva, WI, USA) and the basic performance test of SPID is determined by Oscilloscope (MDO3034, Tektronix, Beaverton, OR, USA). The car is a light truck (D201, Changan, Chongqing, China).

## 4. Conclusions

A self-powered intelligent device based on a piezoelectric generator was designed. The SPID included a piezoelectric power generation unit and an LED warning unit or sensor unit. The SPID was installed on the automobile exhaust device to provide sensing and warning functions by collecting vibration energy. The SPID combined a piezoelectric generator and sensor unit. Testing showed that the overall piezoelectric generator unit could work at different amplitudes and frequencies, and the output power of a single PZT unit was approximately 6.6 μW. Series and parallel tests showed that an output power of 23.4 μW could be achieved when four generating units were connected in parallel. Additionally, a demonstration experiment revealed that the piezoelectric generator could effectively drive 35 LED lamps. Alternatively, commercial temperature and humidity sensors could be driven by commercial capacitors and rectifier bridges, demonstrating that the piezoelectric power generation unit was fully capable of driving intelligent sensors, and the generated electrical signals reflected the operating state of the machine and detected the operating conditions of the machine. The above experiments demonstrated that the intelligent device could be used in building an intelligent sensor system outside the vehicle to recover the vibration energy of the vehicle exhaust system and monitor the driving state. It may also be used in intelligent vehicles, Internet of Things transportation, and other fields in the future.

## Figures and Tables

**Figure 1 micromachines-14-00491-f001:**
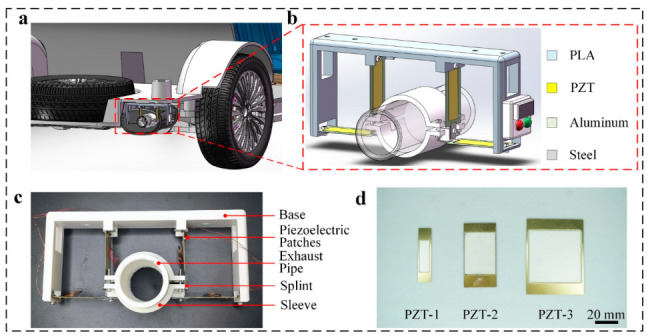
Structural design of the self-powered intelligent device (SPID): (**a**,**b**) simulation assembly, and structure, and (**c**,**d**) physical structure, and three sizes of the piezoelectric plate (scale bar: 1 cm).

**Figure 2 micromachines-14-00491-f002:**
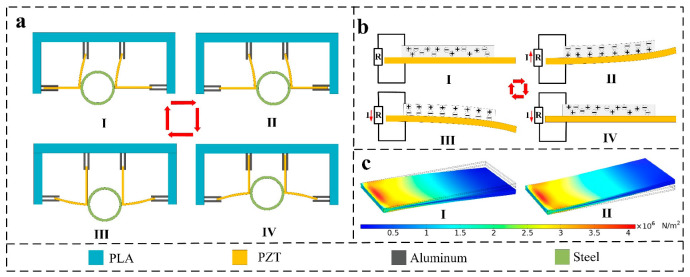
Working principle of the SPID: (**a**) schematic of the operating principle of the intelligent device, (**b**) deformation demonstration and electron distribution of piezoelectric plates, and (**c**) simulated potential distributions of the SPID in the two positional states.

**Figure 3 micromachines-14-00491-f003:**
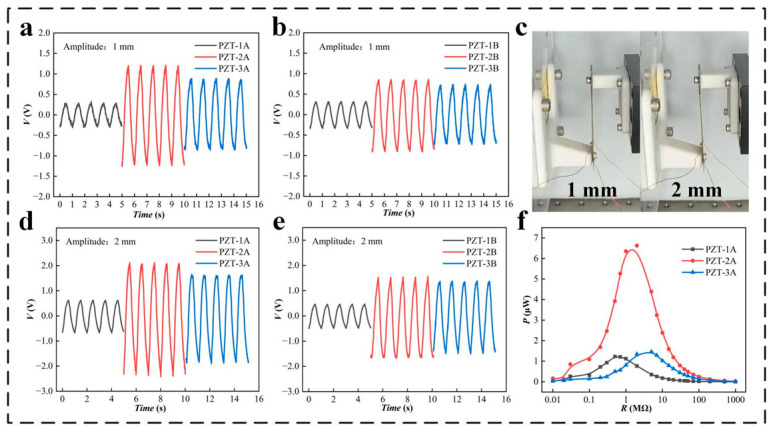
The output performance of the PZT units having different specifications: (**a**,**b**) voltage (*V*) of the PZT units having different specifications for a stroke of the motor of 1 mm, (**c**) photograph of motor strokes of 1 and 2 mm, (**d**,**e**) *V* of the PZT units having different specifications for a stroke of the motor of 2 mm, (**f**) power of the different PZT units for a stroke of the motor of 2 mm.

**Figure 4 micromachines-14-00491-f004:**
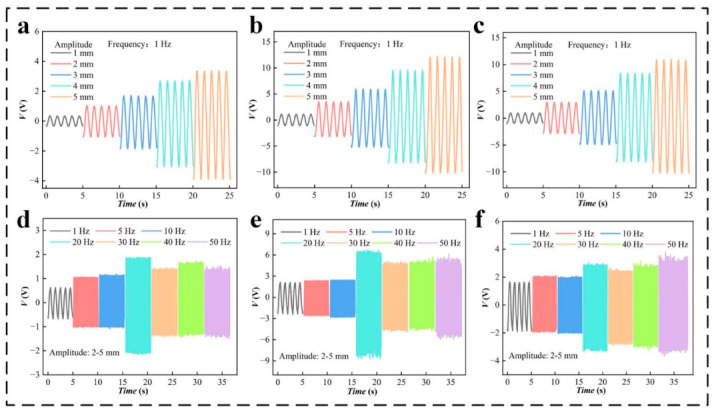
The output performance of PZT units under different amplitudes and frequencies of vibration: (**a**–**c**) voltage (*V*) of PZT units of different specifications for a stroke of the motor of 1, 2, 3, 4, and 5 mm, (**d**–**f**) *V* of PZT units of different specifications for a frequency of excitation of 1, 5, 10, 20, 30, 40, and 50 Hz.

**Figure 5 micromachines-14-00491-f005:**
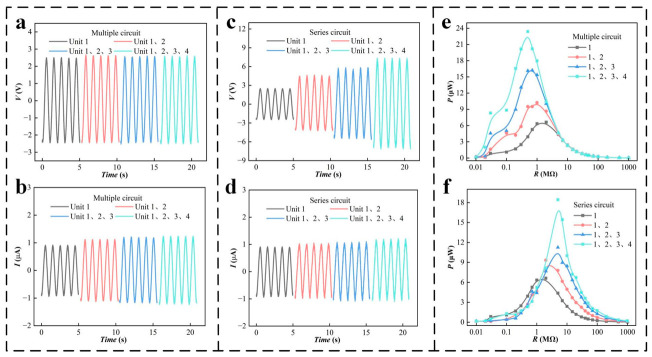
Output performances of parallel and series PZT units: (**a**,**b**) voltage, (*V*) and current, (*I*) output performances of the parallel circuit, (**c**,**d**) *V* and *I* output performances of the series circuit, (**e**,**f**) power output curves of the parallel and series circuits.

**Figure 6 micromachines-14-00491-f006:**
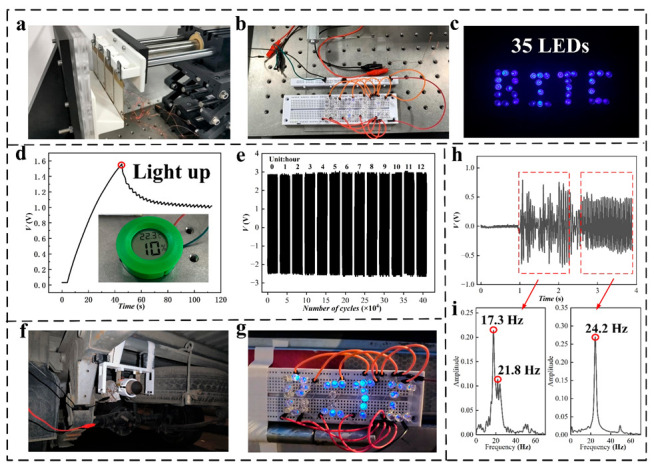
Application experiment of the SPID: (**a**) piezoelectric generator driven by a linear motor; (**b**,**c**) wiring of the LED lamp and photograph after lighting; (**d**) capacitor charging curve and sensor are driven; (**e**) PZT unit operates for 430,000 cycles; (**f**,**g**); intelligent device collecting the vibration energy of the automobile exhaust system and lighting up a warning lamp; (**h**,**i**) piezoelectric generating unit monitoring mechanical vibration signals.

## Data Availability

The data supporting reported results can be made available via requesting the corresponding author.

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
