# Peer review of "Intelligent Device for Harvesting the Vibration Energy of the Automobile Exhaust with a Piezoelectric Generator"

_micromachines, 2023, doi:10.3390/mi14020491_

Round 1

Reviewer 1 Report

In this manuscript, a self-powered intelligent device (SPID) using a piezoelectric energy generator is fabricated for harvesting the vibration energy of the automobile exhaust. The work is very systematic, involving design, mechanism, performance and application. However, there are some minor concerns to be considered before publication. The comments are listed below:

1. In the study of SPID electrical output performance, this paper uses a linear motor to simulate a real vibration environment for relevant tests, and more comparative data should be added for illustration when selecting the best specification piezoelectric sheet, the test results show that PZT-2 piezoelectric sheet has the best output performance, but PZT-3 has the largest piezoelectric material area, can this conclusion be verified by adding theoretical analysis.

2. In this manuscript, the output performance of the PZT cell is tested at different amplitudes and frequencies. The voltage shown in Fig. 4 (a-c) increases gradually with increasing amplitude, why does the frequency curve shown in Fig. 4 (d-f) increase first and then decrease? Please explain the reason for this situation.

3. In this manuscript, the output performance of piezoelectric generators is verified in a comprehensive way. The reviewers suggest that the stability and durability of output performance should be tested in the manuscript or supporting information to make the article more comprehensive and clearer.

4. In this manuscript, it is proposed that the intelligent device can also be used as a vibration sensor and has alarm function. It should have more data to analyze its sensing performance, such as sensor sensitivity, detection range, etc.

5. In addition, the reviewer will also suggest the author to add the latest publications related to energy collection and self-powered sensors used in the field of intelligent vehicles and Internet of Things transportation to the reference section, so as to make the introduction more interesting and comprehensive.

Author Response

We would like to thank the reviewer for bringing up this important point. As the reviewer advised, the manuscript has been revised. Please see the attachment.

Reviewer 2 Report

Manuscript title: Intelligent Device for Harvesting the Vibration Energy of the Automobile Exhaust with a Piezoelectric Generator

Manuscript ID: micromachines-2221829

The present manuscript describes a harvesting energy from the automobile exhaust via piezoelectric nanogenerator. After a thorough reading of the MS and considering the journal’s quality, I recommend submitting again after major revision.

Authors are advised to respond to the below-mentioned queries:

1.      The author uses the commercial PZT as piezoelectric nanogenerator for harvesting energy from automobile exhaust, which questioning the novelty of the work, since many report already show the concept of harvesting vibration energy from automobile via piezo and triboelectric nanogenerator. The author need to comment on it?

References:

1)      Sensors and Actuators A: Physical 331 (2021) 112743.

2)      Electrical Energy Systems 30(12) (2020) e12655.

3)      Joule 1 (2017) 480–521.

4)      Integrated Ferroelectrics 141 (2013) 32-44.

2.      It recommends the author to measure a real time electrical output from the automobile exhaust and compare their result to the lab scale electrical output.

3.      The stability of the energy harvesting from real time need to be provided in the revised MS.

4.      It recommends the author to measure an electrical output at different temperature, since automobile exhaust creates heat up to 400 to 500 F.

5.      Th current, peak power, load resistance profile, capacitor charging (for LED application), rectified voltage and current profile of the nanogenerator is missing in the MS.

6.      The simulation software name and what parameter they used for simulation is missing in the MS.

7.      The introduction is too general, and recently developed new materials are not addressed in this section. I recommend the authors to discuss the usefulness of the PZT in more detail than that of the newly developed materials using the given literature.

References:

1)      Nano Energy 106 (2023) 108096.

2)      ACS Appl. Mater. Interfaces 13 (23) (2021) 27245–27254.

3)      Nano Energy 59 (2019) 453-463.

4)      Matter 5(4) (2022) 1296-1304.

8.      In most of the MS, the continuity is improper. An author needs to recheck it.

9.      The figure quality of the MS can be improved.

Author Response

(The authors gave the same response as above.)

Reviewer 3 Report

(1)   Page 2, Line 60: ‘in view of the limitations of current research on the collection of energy from exhaust system…….’. The paper should review the current research on energy harvesting from exhast system and identify the limimations. However, this has not been done at all and therefore, the reviewer cannot see the limiations of current research and the novelty of the paper at all.

(2)   Regarding how the device works. From the paper, the PZT works in a bending mode to generate electricity. In the reviewer’s understanding, if the pipe is moving downward, the two vertical PZT will prevent the piple from doing so. Therefore, the two horizontal PZT will not be bended. If the pipe is moving from left to right, the left horizontal PZT will prevent the pipe from doing so. The only movement that can generate bending in the PZT is moving upward. The paper should explain clearly how the device works.

(3)   Figure 2 b & c are trival. So does the description of the how piezoelectric material generates electricity in Page 3.

(4)   Page 4. Performance of side A is better than side B, why? It looks strange to the reviewer.

(5)   Page 4, line 127, the overall voltage at a stroke of 2 mm is higher than that at a stroke of 1 mm. This is obvious and the sentence does not add any value to the paper.

(6)   Why PZT 2 has the higherst perfomance? Why not PZT3 which has a larger dimension than PZT2?

(7)   Testing series and parallel connection of PZT sheets. Why the parallel connection outputs a higher power? Theoretically, the connection shouldn’t affect the power output.

(8)   The experiemnt used amplitude of 1-10 mm and frequencies up to 50 Hz. Can the paper justify the use of these conditions? Are they arbitarily determined or are they based on any real-world measurement/estimation?

(9)   The paper used four PZT units assembed on a linear motor to do the demonstration. Why not assemble the PZT as it would work in reality and test the performance? In the current tests, the four PZT will be deflected in the same magnitude which is unlikely to happen in reality.

(10)Most figures in Figure 6 do not provie useful information.

Author Response

(The authors gave the same response as above.)

Round 2

Reviewer 2 Report

The authors have addressed all the issues arised during the peer review time. Therefore I could recommend this article for publication in its current form.

Author Response

We would like to express our heartfelt thanks to the reviewers again.

Reviewer 3 Report

Thanks for the authors reply. My further comments are below.

1.       Line 61: in view of less research on……. This sentence should be ‘in view of little research on….’ 

2.       Line 58: The automobile exhaust system has a large area of vibration [47-49]. The reference [49] looks like having nothing to do with automibile exhaust system. The authors should only use relevant references in the paper.

3.       The piezoelectric structure used is pizoelectric bimorph. Why not use both sides of the bimorph but use only one side, although Side A has a better performance than Side B?

4.       Regarding the series and parallel connection. PZTs connected in series produce a larger voltage output but also have a larger internal resistance than PZTs connected in paralle. Therefore, PZTs in series produce high voltage and low current while PZTs in parallel produce low voltage high power. However, the output power should be the same. This is common sense in piezoelectric energy harvesting. The authors explaination does not make sense to the reviewer.

5.       Regarding the testing condition 25-60 Hz and 1-5 mm. The author should put them in the paper with proper reference to the source of the data.

Author Response

(The authors gave the same response as above.)

Round 3

Reviewer 3 Report

Comments are addressed